# Clustering predicted structures at the scale of the known protein universe

Inigo Barrio-Hernandez[1,8], Jingi Yeo[2,8], Jürgen Jänes[3], Milot Mirdita[2], Cameron L. M. Gilchrist[2], Tanita Wein[4], Mihaly Varadi[1], Sameer Velankar[1], Pedro Beltrao[3,5 ✉] & Martin Steinegger[2,6,7 ✉]

Proteins are key to all cellular processes and their structure is important in understanding their function and evolution. Sequence-based predictions of protein structures have increased in accuracy[1], and over 214 million predicted structures are available in the AlphaFold database[2]. However, studying protein structures at this scale requires highly efficient methods. Here, we developed a structural-alignment-based clustering algorithm—Foldseek cluster—that can cluster hundreds of millions of structures. Using this method, we have clustered all of the structures in the AlphaFold database, identifying 2.30 million non-singleton structural clusters, of which 31% lack annotations representing probable previously undescribed structures. Clusters without annotation tend to have few representatives covering only 4% of all proteins in the AlphaFold database. Evolutionary analysis suggests that most clusters are ancient in origin but 4% seem to be species specific, representing lower-quality predictions or examples of de novo gene birth. We also show how structural comparisons can be used to predict domain families and their relationships, identifying examples of remote structural similarity. On the basis of these analyses, we identify several examples of human immune-related proteins with putative remote homology in prokaryotic species, illustrating the value of this resource for studying protein function and evolution across the tree of life.

Proteins are the major actors in all cellular processes, from the generation of energy to the division of the cell. Knowing their structure is relevant for studying their function, their evolution and potentially for the design of drugs. Although our knowledge of protein sequences has grown considerably over the past years, reaching over hundreds of millions of sequences, the knowledge of their 3D structures has lagged behind owing to the lack of highly scalable experimental methods. Improvements in methods for predicting structure from sequences[1,3,4] now enable the scalable prediction of protein structures for the known protein universe. The AlphaFold Protein Structure Database (AFDB) is a publicly available data repository of protein structures and their confidence metrics, predicted using the AlphaFold2 AI system[1,2]. The AlphaFold-predicted structures have been generally assessed to be of high quality when the predicted local distance difference test (pLDDT) confidence metrics are accounted for, despite remaining inferior to experimentally determined structures[5]. AlphaFold2 and its predicted structures have now been used for diverse applications, including studies of protein pockets[6], prediction of structures of complexes[7,8], studies of structural similarity[9], novel fold predictions[10] and even improvement of genomic annotation[11].

The large increase in available predicted protein structures has spurred the development of more efficient computational approaches, including structural data file compressions[12], methods for pocket predictions[13,14] and comparison of protein structures through structural alignments. For the latter, Foldseek has been developed. Foldseek can increase the speed of comparisons of structures by four to five orders of magnitudes relative to previous approaches while maintaining sensitivity[15], making it possible to perform structural comparisons at a large scale. Clustering proteins by their structure is a crucial tool for analysing structural databases as it enables the grouping of remotely related proteins. Identifying distant relationships might provide valuable insights into protein structure evolution and function. For example, protein family analysis of the initial release of about 365,000 structures[10,16], covering the proteomes of humans and 20 model organisms, suggested that 92% of predicted domains within this set match existing domain superfamilies. However, comparing all 214 million structures against each other using current methods would take approximately 10 years on a 64-core machine. To speed up the process of clustering amino acid sequences, a linear time algorithm, Linclust[17], has been proposed to reduce the computational time significantly. However, such methods have yet to be applied to clustering by protein structural similarity.

Here, we analysed the AlphaFold Protein Structure Database, which contains predicted structures for 214 million proteins across the tree of life. To be able to explore this resource, we developed a highly scalable structure-based clustering algorithm based on Linclust[17] (Methods and Extended Data Fig. 1) that structurally aligns and clusters 52 million structures in 5 days on 64 cores. We clustered the AlphaFold structural

[1]European Molecular Biology Laboratory, European Bioinformatics Institute (EMBL-EBI), Wellcome Genome Campus, Cambridge, UK. [2]School of Biological Sciences, Seoul National University, Seoul, South Korea. [3]Department of Biology, Institute of Molecular Systems Biology, ETH Zurich, Zurich, Switzerland. [4]Department of Molecular Genetics, Weizmann Institute of Science, Rehovot, Israel. [5]Swiss Institute of Bioinformatics, Lausanne, Switzerland. [6]Artificial Intelligence Institute, Seoul National University, Seoul, South Korea. [7]Institute of Molecular Biology and Genetics, Seoul National University, Seoul, South Korea. [8]These authors contributed equally: Inigo Barrio-Hernandez, Jingi Yeo. ✉e-mail: beltrao@imsb.biol.ethz.ch; martin.steinegger@snu.ac.kr

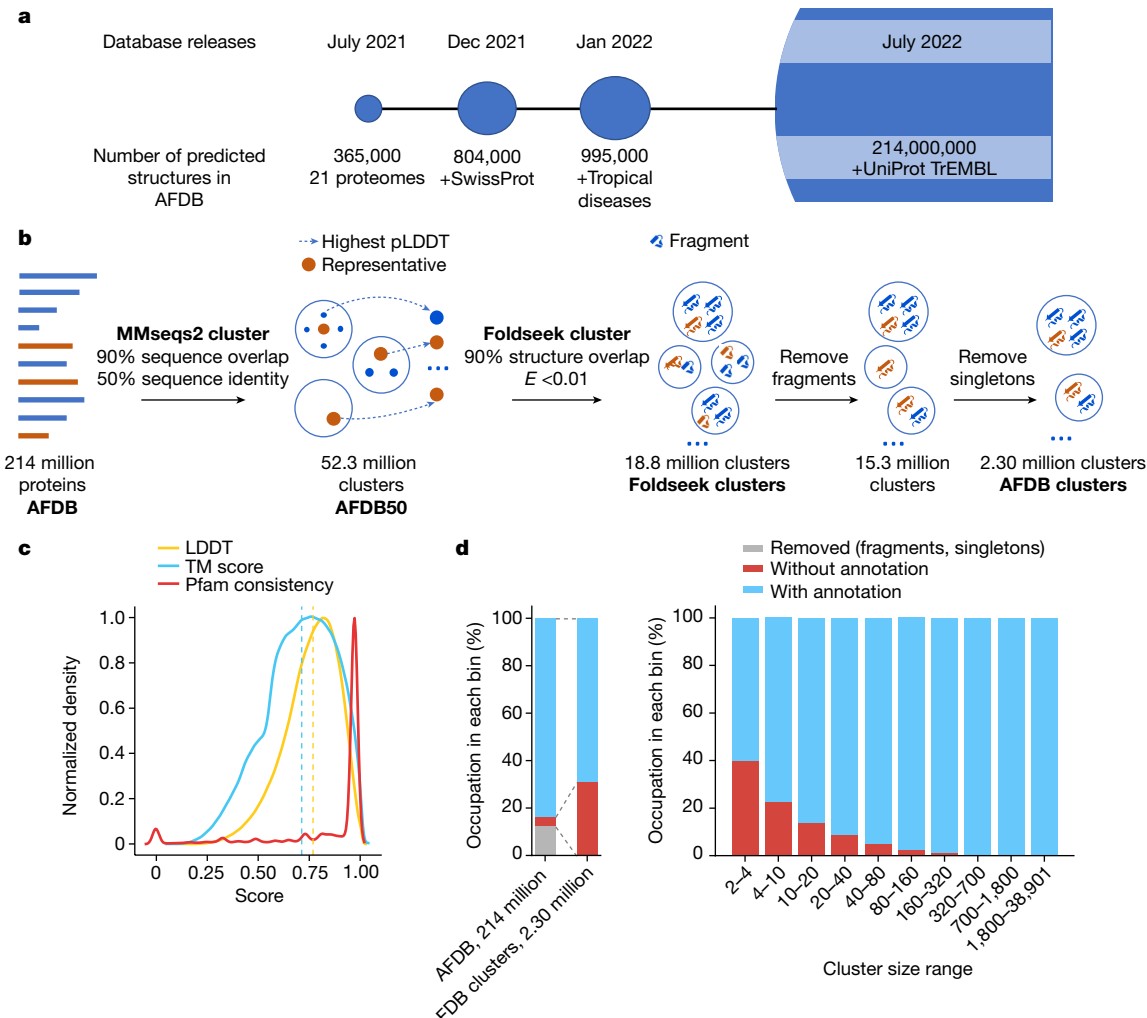

**Fig. 1 | The AFDB, structural clustering workflow and summary of the clusters. a**, The AFDB started as a collaborative effort between EMBL-EBI and DeepMind in 2021. The database grew in multiple stages, with the latest version of 2022 containing over 214 million predicted protein structures and their confidence metrics. **b**, A two-step approach was used to cluster proteins in the database. First, MMseqs2 was used to cluster 214 million UniProtKB protein sequences (AFDB) on the basis of 50% sequence identity and 90% sequence overlap, resulting in a reduction of the database size to 52 million clusters (AFDB50). For each cluster, the protein with the highest pLDDT score was selected as the representative. Next, using Foldseek, the representative structures were clustered into 18.8 million clusters (Foldseek clusters) without

a sequence identity threshold, but still enforcing a 90% sequence overlap and an *E*-value of less than 0.01 for each structural alignment. As the last step, we removed all sequences labelled as fragments from the clustering, ending up with 2.30 million clusters with at least two structures (AFDB clusters). **c**, AFDB cluster structural and Pfam consistency. Our clusters have a median LDDT of 0.77 and a median TM score of 0.71 across all clusters and 66.5% of clusters with Pfam annotations are 100% consistent. **d**, Summary of sequences and clusters with and without annotation (left) and the relationship of cluster sizes to annotation (right). From left to right, each bin occupies AFDB clusters at rates of 12.24%, 10.59%, 9.20%, 10.07%, 10.46%, 10.05%, 9.04%, 9.20%, 9.19% and 9.96%.

database into 2.30 million clusters with 31% of clusters—representing 4% of protein sequences—not matching previously known structural or domain family annotations. We found that 532,478 clusters have representatives present in all of the tree of life and we found several species-specific structural clusters that could contain examples of de novo gene birth events. Finally, we used structural comparisons to predict domain families and their relationships identifying putative remote homologies that expand the evolutionary coverage of previously known families.

## Structure-based clustering of the AFDB

The AFDB covers over 214 million predicted protein structures and has grown in several stages (Fig. 1a). The initial release focused on 20 key model organisms, while subsequent updates provided predictions for the Swiss-Prot dataset of the Universal Protein Resource[18]

(UniProt) and proteomes relevant to global health, taken from priority lists compiled by the World Health Organisation. The current update covers most of the TrEMBL dataset of UniProt. The AFDB parses and archives these data and makes them accessible through bulk download options, programmatic access end points and interactive web pages. The programmatic access, in particular, facilitated the integration of AlphaFold models into other biological data repositories, such as Protein Data Bank Europe (PDBe)[19], UniProt[18], Pfam[20], InterPro[21] and Ensembl[22].

To gain insights into the 214,684,311 structures of the AlphaFold UniProt v.3 database we developed a scalable clustering approach in two steps as depicted in Fig. 1b. The first step involved using MMseqs2 (ref. 23) to cluster the database on the basis of 50% sequence identity and a 90% sequence alignment overlap of both sequences, resulting in 52,327,413 clusters. For each cluster, the protein structure with the highest confidence (that is, the highest pLDDT score) was

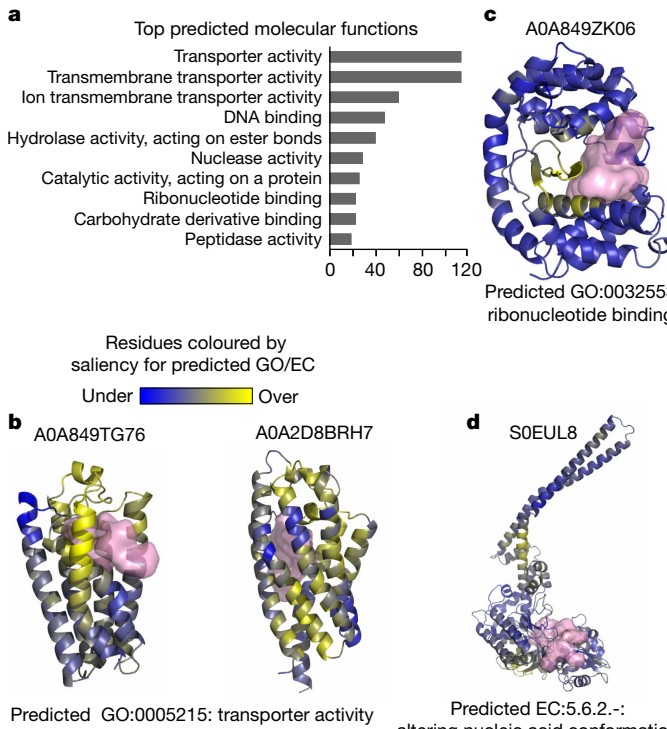

**a** Top predicted molecular functions

Transporter activity
Transmembrane transporter activity
Ion transmembrane transporter activity
DNA binding
Hydrolase activity, acting on ester bonds
Nuclease activity
Catalytic activity, acting on a protein
Ribonucleotide binding
Carbohydrate derivative binding
Peptidase activity

0    40    80    120

**c** A0A849ZK06

Predicted GO:0032553
ribonucleotide binding

Residues coloured by
saliency for predicted GO/EC

Under ▮▮▮▮▮ Over
(blue)        (yellow)

**b** A0A849TG76    A0A2D8BRH7

Predicted GO:0005215: transporter activity

**d** S0EUL8

Predicted EC:5.6.2.-:
altering nucleic acid conformation

**Fig. 2 | Putative novel enzymes and small-molecule-binding proteins in structures lacking annotation. a**, Counts of GO molecular function terms that are most often predicted by DeepFRI on the set of selected 1,707 structures with predicted pockets. **b**–**d**, Examples of structures (A0A849TG76 and A0A2D8BRH7 (**b**), A0A849ZK06 (**c**) and S0EUL8 (**d**)) with predicted pockets and functional annotations. Each example shows the UniProt ID (top), the highest-scoring DeepFRI function prediction (bottom) and the top-scoring pocket (pink surface). The structures are coloured by residue-level contributions to the DeepFRI function predictions, ranging from blue (no contribution) to yellow (strong contribution).

selected as the representative. Clustering proteins by structural similarity remains computationally intensive and difficult to scale. For this reason, we developed a structure-based clustering algorithm based on Foldseek (Methods). In brief, we adapted Linclust and MMseqs2 sequence clustering algorithms to the three-dimensional interaction (3Di) structural alphabet used in Foldseek to allow structural clusters in linear time complexity. Our structural clustering method resulted in the identification of 18,661,407 clusters, using an *E*-value of 0.01 and structural alignment overlap of 90% of both sequence criteria. It took 129 h on 64 cores to finish the clustering. As the final step, we removed every sequence labelled as a 'fragment' in UniProt. This identified 2,302,908 non-singleton clusters that have on average 13.05 proteins per cluster with an average pLDDT of 71.64. The remaining 13,012,338 singleton clusters have an average pLDDT of 58.95.

## Cluster purity analysis

We measured the quality of our AFDB clusters (Fig. 1c) by assessing their structural and Pfam consistency. By aligning each cluster member to a representative, we found that the clusters tend to be structurally homogeneous as judged by two structural similarity metrics (median LDDT of 0.77 and a median template modelling (TM) score of 0.71; Fig. 1c). Similarly, we found that members of the same clusters tend to have the same Pfam domain (Fig. 1c), with 66.4% having 100% consistency. The relationship between cluster members and consistency score (Extended Data Fig. 2a) reveals that clusters even with thousands of members have perfect consistency. We further examined the

relationship between structural and functional similarity using Pfam (Extended Data Fig. 2b) and Enzyme Commission (EC) annotations (Extended Data Fig. 3). As anticipated, an increase in LDDT corresponds to an increase in functional similarity. With increasing LDDT from 0.5 to 0.9, there is an increase in the percentage of perfectly consistent clusters—from 29.2% to 93.8% in Pfam, and from 40.3% to 81.3% for EC (level 4), respectively.

Structural similarity between two sequences can often be traced back to either shared evolutionary roots (homologues), or it can be a result of convergent evolution (analogues). We therefore investigated the evolutionary relationships within clusters produced by our method using the Evolutionary Classification of Protein Domains (ECOD) database[24]. This hierarchical domain database delineates the evolutionary relationships between protein domains. This analysis showed that 97.4% of pairwise compared cluster members are conserved at the H-group (homology) level (Methods). This analysis suggests that our clusters are probably composed primarily by homologues, although specific examples will require further evolutionary analysis.

## Clusters of unknown structure and function

The availability of predicted structures covering a large fraction of the known protein universe enables us to examine what fraction of this structural space is novel. We tried to uncover structurally and functionally unknown protein clusters in the AFDB dataset—defined as 'dark clusters'. We first identified 1,135,118 (49% of AFDB clusters) clusters that were found to be at least partially similar to previously known structures in the PDB (Methods). The representative proteins of the remaining clusters were next annotated to the Pfam database by MMseqs2 search, resulting in 883,788 (38% of AFDB clusters) dark clusters (Methods). Finally, we identified clusters containing members with Pfam or TIGRFAM[25] annotations in the UniProt/TrEMBL and Swiss-Prot database. This resulted in the identification of 711,705 (30.9% of AFDB clusters) dark clusters, probably enriched for novel structures.

The distribution of the known and unknown clusters as a function of their size is shown in Fig. 1d. The sizes of clusters that lack annotations are smaller compared with the annotated clusters. For this reason, the dark clusters map to a proportionally smaller fraction of the protein universe. Although these clusters comprise approximately 30.9% of the AFDB clusters, they represent only 4.06% of the AFDB. This is consistent with the expectation that structures with many representatives in the protein universe are better studied and that the vast majority of protein structures can be annotated with at least partial similarity to a known structure of domain family annotation.

## Novel enzymes and small-molecule binders

From the 711,705 clusters without annotations (dark clusters), we selected 33,842 clusters with the highest average AlphaFold2 prediction confidence (that is, average pLDDT >90). For each, we picked the member with the highest confidence for further investigation. To predict potential novel enzymes, we searched each structure for pockets and predicted Gene Ontology (GO) and EC number using DeepFRI, a structure-based function prediction method (Methods). In total, we identified 1,770 pockets in 1,707 structures and made 5,324 functional assignments within these proteins with predicted pockets. The pocket prediction led to the identification of high-confidence structure predictions (pLDDT >90) that do not appear to be correct. From 1,770 pockets, 579 (32.7%) encompass more than 40% of the total protein sequence, indicating that the predicted structure is not compact. Manual inspection of these structures (examples are shown in Extended Data Fig. 4) confirmed this lack of compactness and secondary structural elements. We hypothesize that several of these are probably incorrect predictions.

The top most often predicted molecular functions are shown in Fig. 2a with the top three including the term 'transporter activity'.

Similarly, the most often predicted cellular component was 'intrinsic component of membrane' (379 annotations). This indicates that structures without annotations may be enriched for membrane-bound proteins that have been historically difficult to determine experimentally. This is also the case when considering all 711,705 dark clusters predicted by DeepFRI (Extended Data Fig. 5). Two examples of putative transporters are shown in Fig. 2b, including the top predicted pocket and coloured by the residue importance given by DeepFRI for this predicted function. In addition to the putative transporters, there is a wide diversity of other predicted functions. For example, UniProt A0A849ZK06 (Fig. 2c) is predicted to be a ribonucleotide-binding protein with an overall structure having an organization that resembles a protein kinase fold. The residues contributing the most to the DeepFRI prediction are directly in contact with the top scoring pocket (Fig. 2c), suggesting a potential nucleotide-binding function for this pocket. Finally, UniProt S0EUL8 (Fig. 2d) has a top prediction of EC 5.6.2.-, which annotates enzymes that can alter nucleic acid conformations. The structure resembles members of the structural maintenance of chromosomes family but it is missing several characteristic elements. The preceding gene in the genome encodes a RecN homologue (a member of the structural maintenance of chromosomes family), giving additional evidence for a role of UniProt S0EUL8 in chromosome maintenance.

## Taxonomic analysis of the clusters

To gain insights into the distribution of the identified structural clusters, we examined their taxonomic composition to determine the extent of protein machinery shared across different super-kingdoms (Fig. 3a). For this, we mapped the members of the cluster in the tree of life and identified the most recent common ancestor for all members of the cluster (Methods). In this way, we mapped non-singleton structural clusters that appear to be conserved at the cellular organism (23%) (that is, universal to all life), bacterial (16.1%), Eukaryota (13.5%) and Archaea (0.5%) levels. Together, this suggests that the majority of the structural clusters are probably very ancient in origin.

Although the majority of protein clusters is mapped to the common ancestor of eukarya or older, we found a small fraction (3.91%) of species-specific structural clusters. Compared with other clusters, the species-specific clusters tend to have fewer members (that is, twice more likely to have just two members); they are more likely to be dark, with 56% having no annotation; and composed of smaller proteins, with a median length of around 40 amino acid fewer). However, the overall prediction confidence (pLDDT) of the species-specific clusters is comparable to that of the remaining clusters, with an average of 69.35 compared to 71.73. The organisms with the largest species-specific clusters are *Acidobacteria bacterium*, *Araneus ventricosus*, *Escherichia coli*, *Sepia pharaonis* and *Chloroflexi bacterium*, which range from 1,884 to 1,390 clusters.

## Human-related cluster analysis

As an example application, we studied human protein-containing clusters from an evolutionary conservation perspective. We mapped the clusters containing human proteins to the tree of life (Extended Data Fig. 6) and first looked for human-specific clusters (that is, containing only human proteins). Out of the 13 human-specific clusters identified, 9 are predicted non-confident with a pLDDT score of less than 70 and did not contain structural proteins. The remaining four clusters contained a herpes virus U54 (UniProt: A0A126LB04) unit; annexin (UniProt: A0A4D5RA95) with limited human homologues in UniRef50; a U2 snRNP-specific A' protein (UniProt: Q9UEN1) that appeared to be a fragment but is not labelled as one; and VPS53 (UniProt: A0A7P0T9Z7), a single long coil structure that was not clustered by Foldseek due to high random chances of observing such a structure. Our findings do

not support the presence of newly emerging human-specific structural clusters within the set of human sequences annotated in UniProt. However, this does take into account singleton clusters.

We next extracted all clusters containing a human protein and associated each human cluster with its corresponding GO terms and lowest common ancestor (LCA). When multiple human sequences were present in a cluster, the GO annotation of the human protein with the highest pLDDT score was selected. A small selection of GO annotations that highlight the evolutionary conservation of human structures is shown in Fig. 3b. Human proteins with similar structures across most of the tree of life are annotated with a diverse set of terms including several enzyme activities (for example, ligase activity, oxidoreductase activity, serine-type endopeptidase activity). Present in bacteria and eukarya, proteins linked with the microtubule-organizing centre and voltage-gated potassium channel activity are included. Mostly restricted to eukarya, terms such as nucleus, chromatin organization and microtubule motor activity are included. More recently evolved structures include annotations such as immune response and hormone activity.

## Bacterial and human immunity protein links

Note that, even if some biological processes were primarily restricted to eukarya or more recently diverged clades, we could find cluster representatives that were present in bacterial species. For example, most human proteins that are annotated to the nucleus (GO:0005634) are in clusters mapped to eukarya as their LCA. However, we found exceptions including, for example, a histone-related cluster (Fig. 3c) supporting the previously reported evolutionary link between eukaryotic and bacterial histones[26]. Similarly, we found several immunity-related proteins with structural similar proteins present in bacteria. These include TNFRSF4 (UniProt: P43489) with similar structures in bacteria due to common cysteine-rich repeat regions that overlap with the TNFR/NGFR cysteine-rich region domain annotations in InterPro (IPR001368). We also found bacterial structures that are related to the human CD4 like protein B4E1T0 (Extended Data Fig. 7a), although these can also be annotated by sequence matching to the immunoglobulin-like domain family in InterPro (IPR013783).

The structural similarity between human and bacterial proteins may also inform on their function in bacteria. The human bactericidal permeability-increasing (BPI) protein (B4DKH6) is a key component of the innate immune system and is known to have a strong affinity for negatively charged lipopolysaccharides found in Gram-negative bacteria. In our analyses, this protein clusters with bacterial structures (Fig 3c), for example, the protein A0A2D5ZNG0, which aligns with the human protein at a TM-score of 0.81 normalized to the length of the human protein. Moreover, searching for partial hits by Foldseek identified that YceB from *E. coli* and other gram-negative bacteria has structural similarity to the C-terminal region of human BPI (Extended Data Fig. 7b). The *E. coli* YceB protein is a tubular putative lipid-binding protein without a well-characterized function. This structural similarity may suggest a role of YceB homologues in regulating the outer membrane.

Our analysis identified a cluster containing the human protein AIM2 (O14862), which recognizes pathogenic double-stranded DNA[27] and leads to the formation of the AIM2 inflammasome. When searching the NR database using NCBI BLAST[28], we found no bacterial hits for the human *AIM2* gene. However, three structures in '*Candidatus* Lokiarchaeota archaeon' and one in the bacterium *Clostridium* sp. from an uncultured source (UniProt: A0A1C5UEQ5) were identified as similar to human AIM2 in our analysis. The bacterial protein (UniProt: A0A1C5UEQ5), encoded on a contig of length 138,559 (GenBank: FMFM01000010), is unlikely to be a contaminant due to its length[29]. UniProt A0A1C5UEQ5 is not unique, as many homologous sequences, mostly labelled as 'hypothetical protein', were found in the NR database

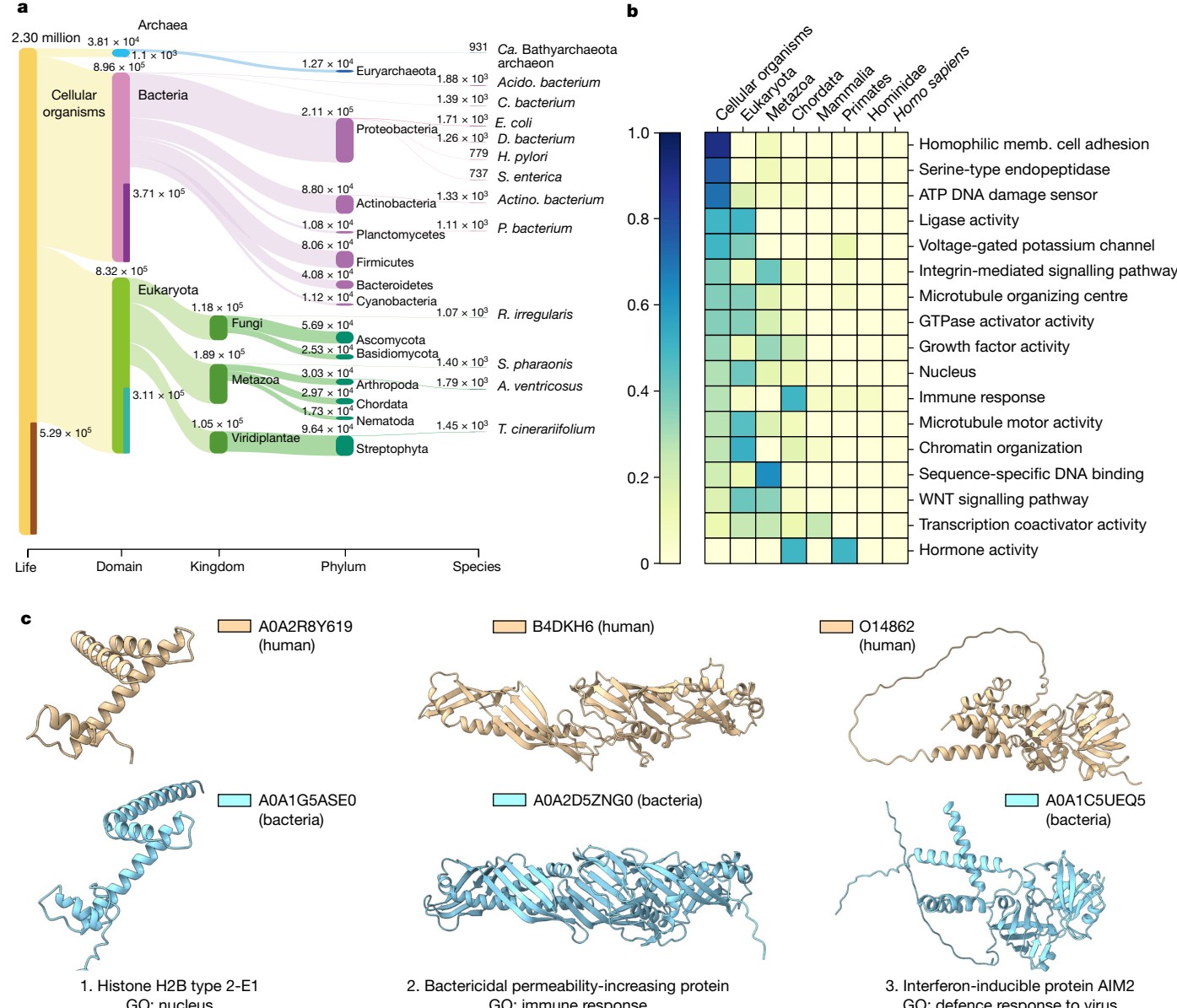

**Fig. 3 | Evolutionary distribution of clusters and human-centric cluster analysis. a**, Visualization of the LCA of all non-singleton clusters as a Sankey plot produced by Pavian. Only the largest 13 taxonomical nodes per rank are shown. **b**, The distribution of selected GO terms across the human lineage of the LCA based on the analysis of human protein-containing clusters (abundance is normalized per GO category). **c**, Three example structures from the human clusters that are conserved across humans and bacteria, among the eukaryote GO-annotated clusters. A histone protein with a nucleus GO annotation, which was found to be conserved at the cellular organism level and supports the previously reported evolutionary connection between eukaryotic and bacterial

histones (left)[26]. The human innate immunity genes *BPI* (middle) and *AIM2* (right) encode structurally similar proteins in bacterial species, highlighting the potential for cross-kingdom sharing of immunity-related proteins. *Acido. bacterium, Acidobacteria bacterium; Actino. bacterium, Actinomycetia bacterium; 'Ca.* Bathyarchaeota archaeon', '*Candidatus* Bathyarchaeota archaeon'; *D. bacterium, Deltaproteobacteria bacterium; H. pylori, Helicobacter pylori;* memb., membrane; *P. bacterium, Planctomycetes bacterium; R. irregularis, Rhizophagus irregularis; S. enterica, Salmonella enterica; T. cinerariifolium, Tanacetum cinerariifolium.*

from mostly uncultured human gut bacterial sequences with higher than 90% sequence identity. We predicted the structure of one homologous protein that is 64% identical to UniProt A0A1C5UEQ5 (Extended Data Fig. 8)—which originates from a cultured *Lachnospiraceae bacterium* that is part of the Culturable Genome Reference[30] of the human gut—using ColabFold[31] and confirmed that it has a similar structure DNA-binding domain structure (TM score of 0.97 and 0.56 in relation to UniProt A0A1C5UEQ5 and human AIM, respectively). These results suggest that the AIM2 inflammasome may have been repurposed from ancient DNA-sensing-related proteins. It is possible that the bacterial versions may also have a role in pathogen DNA sensing and response.

These results exemplify how the structural clusters can provide hypotheses as to the evolutionary origin of specific biological processes and further illustrate the cross-kingdom similarities in immune systems.

## Domain prediction by structure search

The clusters defined above group structurally similar proteins at full length. Proteins are sometimes composed of different regions or domains that can fold independently, with a growing collection of such domain families being catalogued in databases such as Pfam[20]

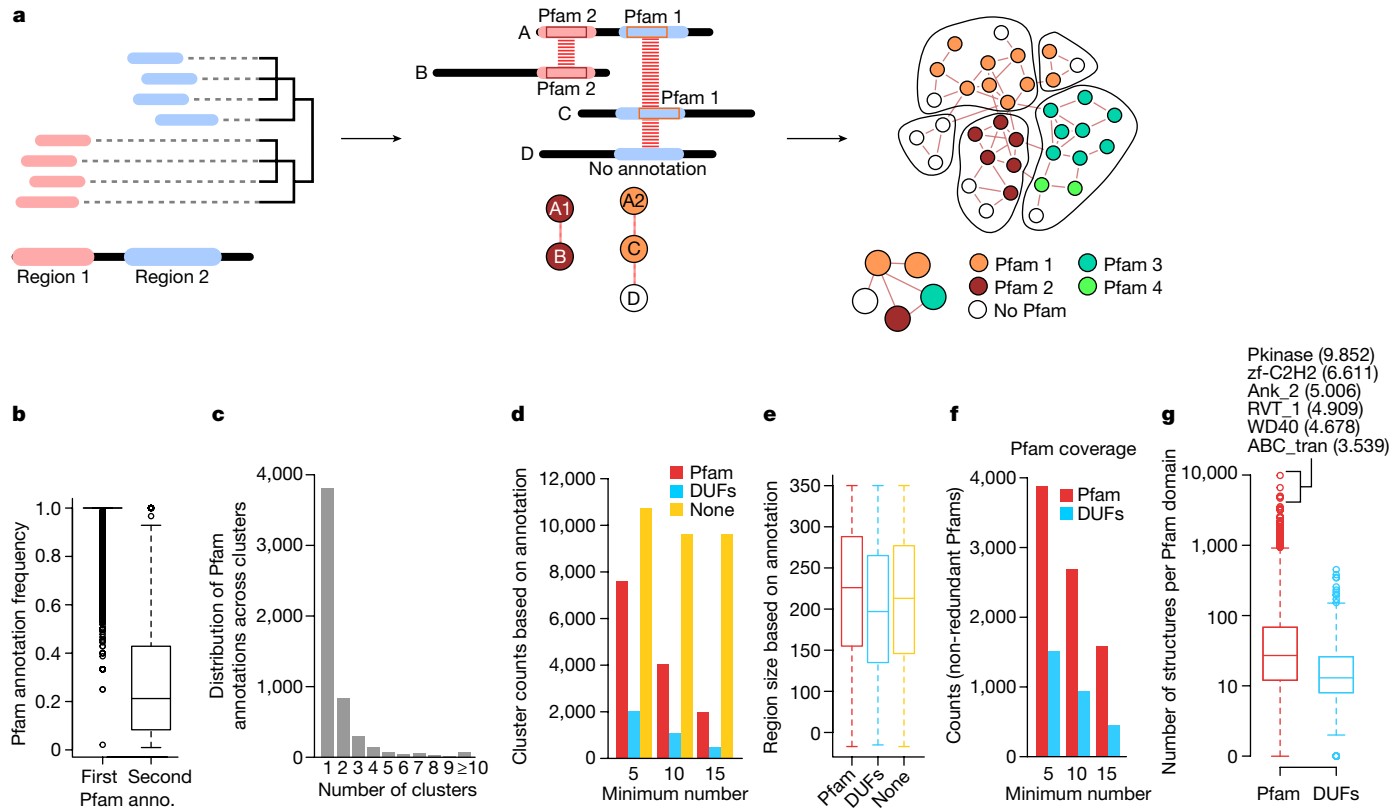

**Fig. 4 | Prediction of domain families by local structural similarity hits.**
**a**, Diagram of the structure-based domain family prediction method. Clustering of the start and end positions for Foldseek hits of one protein against all others was used to define potential domain boundary positions. Each predicted domain region was linked to the others sharing structural similarities and graph-based clustering was used to define domain families and interdomain similarity. **b**, The frequency distribution of the most common (*n* = 9,631) and the second most common (*n* = 1,628) Pfam annotations found members of all predicted domain families. anno., annotation. **c**, The counts of the number of clusters with a given Pfam as the most frequent. **d**, The number of domain family clusters annotated to a Pfam, DUF or no domain annotation. **e**, The distribution of protein region length in the predicted domain families,

stratified by their annotations: Pfam domain (*n* = 1,048,276), DUF (*n* = 72,798) and not annotated (*n* = 1,904,498). **f**, Non-redundant count of Pfam and DUF domain families found in the structure-based predicted families. **g**, The distribution of the number of structures found for each predicted domain family annotated with a known Pfam (*n* = 3,875) or DUF domain (*n* = 1,513). The top 6 Pfam annotations are highlighted using their abbreviations: Pkinase, protein kinase domain PF00069; zf-C2H2, zinc finger, C2H2 type PF00096; Ank_2, ankyrin repeats, PF12796; RVT_1, reverse transcriptase, PF00078; WD40, WD domain, G-beta repeat PF00400; ABC_tran, ABC transporter, PF00005. The box plots in **b**, **e** and **g** show the median (centre line), the quartiles 1 and 3 (box limits) and 1.5 × the interquartile range (whiskers).

or InterPro[21]. Domain family prediction is performed primarily by sequence searches, exploring the fact that domain families have conserved sequence features. The vast increase in protein structures and fast algorithms to compare them opens the possibility of predicting domain families by structural similarity. Here we devised a procedure using structural similarity matches by Foldseek to predict putative domain regions and families (Fig. 4a and Methods). In brief, a representative structure from each of the Foldseek clusters defined above was used for an all-by-all structural similarity search using Foldseek. Although these representative structures should be structurally non-redundant at the full protein level, they will still share many structurally similar domains. For each sequence/structure, we cluster the start and end positions of all Foldseek hits and use these to define probable domain boundaries. The predicted domain regions were then connected if they had structural similarity, and a network clustering method was used to cluster domain regions into putative domain families (Methods).

We used Pfam annotations to assess the quality of these predictions (Fig. 4b–g). For each putative domain family with at least five representatives, we determined the frequency of the first and second most frequent Pfam annotations, with the majority having homogeneous annotations (Fig. 4b). Each Pfam annotation is predominantly

found within a single domain family suggesting that these tend to be non-redundant. For domain families with at least 5 representatives, 7,599 families match Pfam, 2,032 match Pfam domains of unknown function and 10,722 do not match Pfam and are probably enriched in novel families. The median length of the regions is similar for previously known or putative novel families (Fig. 4e). Given that we started with mostly non-redundant structures, we do not expect this approach to recover most domain families. We found 5,388 non-redundant Pfam annotations for predicted domain families with at least 5 representatives, corresponding to around 29% of the 19,000 known Pfam families.

In summary, clustering of local Foldseek hits can accurately predict domain families leading to the prediction of many potential unexplored families. We provide a complete list of all predicted domain families online (https://cluster.foldseek.com/).

## Structural similarity in distant domains

The network clustering procedure used above also enables the identification of pairs of predicted domain families that share some structural similarity. Among such pairs, we found around 500 connections between clusters enriched with a Pfam annotation and other domains

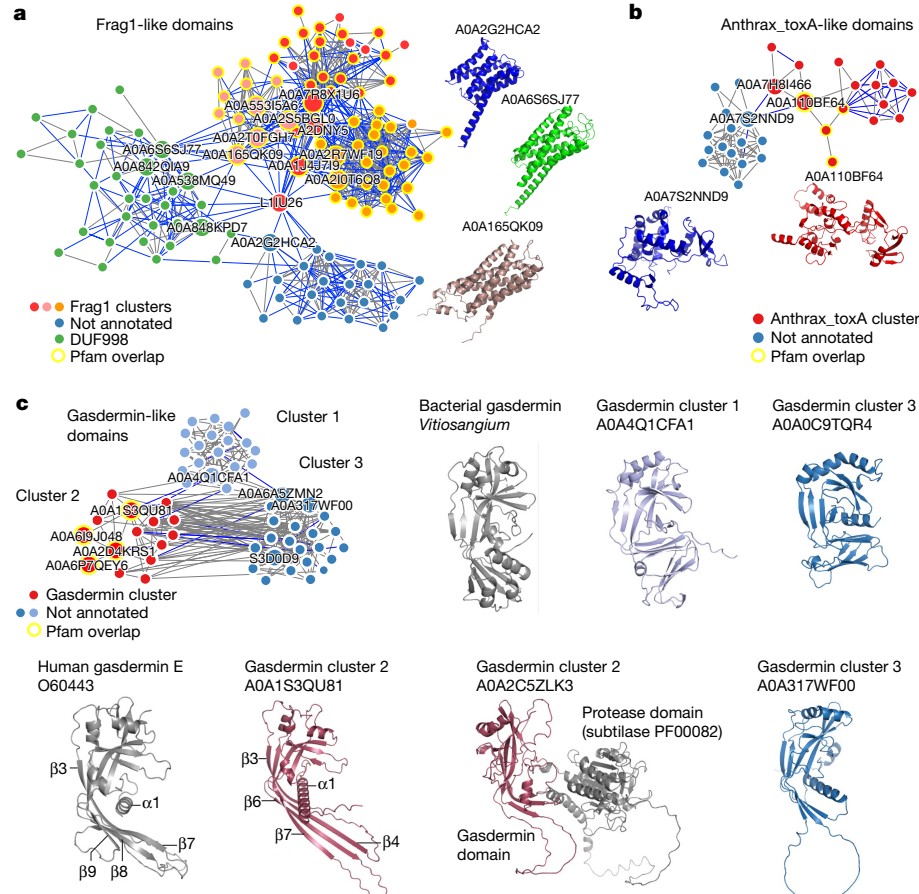

**Fig. 5 | Examples of non-annotated domain families with structural similarity to annotated domain families. a**, Frag1-like domains. Three clusters were found enriched for the Frag1 Pfam annotation that had structural similarity to one cluster enriched for a domain of unknown significance (DUF998) and one cluster without annotations. **b**, Anthrax_toxA-like domains. A cluster enriched for the anthrax_toxA Pfam annotation was found with structural similarity to a cluster with no annotations. **c**, Two clusters without annotations were found with structural similarity to a cluster enriched for the

gasdermin Pfam annotations. Cluster 2 gasdermin N-terminal domain structures reveal homology to human gasdermin E. The corresponding structural characteristics are highlighted. Some gasdermin domains were found fused to protease domains (UniProt: A0A2C5ZLK3). The bacterial gasdermin structure (PDB: 7N51) is similar to novel gasdermin domains from non-annotated cluster 2. The third cluster revealed homology to both animal and bacterial gasdermins.

without clear annotations, providing examples of potential functional annotations. From these, we focused on connected domain families enriched in proteins from different kingdoms (Fig. 5). The Frag1-like domains exemplify the strength of structural-based similarity searching (Fig. 5a). The Frag1/DRAM/Sfk1 Pfam domain (PF10277) annotates proteins with a six-α-helix bundle transmembrane region observed in eukaryotic species. In our analysis, a domain family enriched for this Pfam annotation was linked to two additional families enriched in bacterial and archeal sequences, one enriched for a domain of unknown function (DUF998; PF06197) and a second not annotated. The three families are structurally identical, typically forming a six-α-helix bundle, despite the very low sequence similarity between the sequences forming these.

We also found a cluster enriched for the anthrax_toxA Pfam (PF03497; Fig. 5b), more specifically, the annotated domains contained structures similar to the oedema factor, a calmodulin-activated adenylyl cyclase[32]. The oedema factor is one of the three components forming the bacterial anthrax toxin system. Our analysis identified a structurally similar putative domain family enriched in eukaryotic proteins (Fig. 5b). Specifically, several algae proteins were found to have structures that had partial matches to the oedema-factor-domain-related structures. This raises the possibility that algae might be using similar toxin systems.

## Identification of gasdermin domains

Our search resulted in the identification of two domain families with structural similarity to a cluster enriched for the gasdermin domain (Fig. 5c). In humans, gasdermin is the executor of inflammatory cell death called pyroptosis and is crucial for defence against pathogens. After sensing a pathogen, caspases are activated that cleave off the C-terminal repressor domain of gasdermin, releasing the N-terminal domain to assemble into large pores in the cell membrane[33]. The predicted gasdermin structures from all three groups exhibited the structural characteristic conservation of a twisted central antiparallel β-sheet and the shared placement of connecting helices and strands of gasdermin. The structures enriched in the gasdermin Pfam annotation adopted a similar conformation to that of the mammalian gasdermin N terminus, especially of gasdermin E, which is considered to be evolutionary ancient[34]. In the inactive structure of mammalian gasdermin (A, B, D and E), the N terminus forms interfaces with the repressor C-terminal domain mediating autoinhibition, one of these is the primary interface at the α1 helix[35]. Gasdermin is activated by proteolytic cleavage, which results in N-terminal activation through the lengthening of strands β3, β5, β7 and β8, and oligomerization[36]. Indeed, gasdermin domains from the Pfam annotated group had both the α1 helix as well as the corresponding β-sheets necessary for the active form of gasdermin.

Gasdermin was also recently found in bacteria and archaea, in which it is similarly activated by dedicated proteases and defends against phages by pore-mediated cell death[37]. Notably, the non-annotated group 1 of gasdermin domains displayed strong similarity to the bacterial gasdermin structure (Fig. 5c). The other non-annotated group (cluster 3) showed a large degree of diversity and exhibited features of both mammalian and bacterial gasdermin. In some cases, we observed that the N-terminal gasdermin domain was fused to other domains including proteases (Fig. 5c; UniProt: A0A2C5ZLK3). As gasdermin is activated by proteolytic cleavage, such protein fusion hints at a similar activation mechanism for the novel gasdermin domains.

## Discussion

The orders-of-magnitude increase in available structural models raises challenges in data management and analysis of such large volumes. This difficulty is amplified by the fact that the repository of publicly available structures, consisting of the combined databases of AFDB and the ESMatlas[38], is approaching a billion entries. For this reason, we developed a clustering procedure that can scale to billions of structures, identifying 2.30 million non-singleton clusters of which 31% do not have similarity to previously known structures or domain annotations. These clusters annotate only 4% of protein sequences, indicating that the vast majority of the protein structural space has been at least partially annotated. As the criteria used include partial hits to known structures or domain annotations, the degree of understudied structural space is probably underestimated. As we illustrate, our analysis can guide the prioritization of predicted novel protein families for future computational and experimental characterization.

Structural clustering is a powerful tool for identifying structurally similar proteins that can inform on evolutionary relationships, but its accuracy can be affected by certain limitations. Here we set a 90% alignment overlap as the requirement for assigning a structure to a cluster, which may exclude similar structures with significant insertions or unique repeat arrangements. Moreover, our strict *E*-value threshold of 0.01 may result in missed similarities. Another limitation is that the current AFDB does not contain the full extent of protein sequences from metagenomics studies or viral proteins, limiting the potential to detect retroviral proteins.

In addition to the full-length protein clustering, we used Foldseek's local hit matches to predict and cluster protein regions into putative domain families. The protein region clusters tend to overlap well with previous definitions of domain families as annotated in the Pfam database and led to the identification of over 10,000 unassigned domain-level clusters that should be enriched in putative novel domain families. We did not perform exhaustive searches with other sequence-based domain family annotations that could annotate additional clusters with previous knowledge. Note that we considered only the representatives of Foldseek clusters when performing the domain prediction. As the domain prediction requires multiple observations on the same structural region, additional domains are expected to be detected if each structure was searched against a larger set of structures.

As protein structure is conserved for longer periods of evolutionary time than protein sequences, we expect that AFDB will empower the identification of remote homology. Although some advanced sequence-based methods can already assist in this task[39–41], the availability of predicted structures may help identify meaningful evolutionary relationships. From an analysis of curated protein families, we find that our clusters are enriched preferentially in homologous over analogous relationships (Methods). Nevertheless, one should still be cautious when interpreting structural similarity as evolutionary homology. Our analysis here provides several examples of structural similarity across kingdoms that is indicative of remote homology. In particular, we focused on several examples relating human immunity to bacterial structures, emphasizing how some ancient systems have been co-opted for use in the mammalian immune response system. We expect that many more examples can be derived from the clustering results provided here.

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

## Methods

### Structural clustering algorithm

The clustering procedure is similar to MMseqs2's clustering but, instead of using sequences, Foldseek's 3Di alphabet (Extended Data Fig. 1) was used to represent the structures as one-dimensional sequences. The clustering algorithm combines Linclust[17] and cascaded MMseqs2 (ref. 42) clustering. The pipeline applies this strategy to allow for efficient clustering of millions of structures. First, protein structures are converted to 3Di sequences and processed according to the Linclust workflow. This includes extracting $m$ $k$-mers (default $m = 300$, $k = 10$) from each sequence and grouping them on the basis of their hash value. The $k$-mer groups are then used to assign each structure to the longest sequence (representative) within the group. The shared diagonal on which the $k$-mer is found is also stored for further use in the alignment step.

The pipeline then proceeds with an ungapped alignment algorithm that rescores the structures on the basis of the shared diagonal between members and representatives using 3Di and amino acid information. The sequences that meet the defined alignment criteria, such as $E$-value, alignment coverage, sequence identity, alignment LDDT[43] or TM score[44], are clustered using the MMseqs2 clustering module (default using the set-cover algorithm). After this step, the structures that have been assigned already are removed from the set and the remaining representative member hits are aligned using Foldseek's structural Gotoh–Smith–Waterman algorithm[15], and all passing hits are clustered as well. The remaining cluster representatives are successively clustered by three cascaded steps of prefiltering, structural Smith–Waterman alignment and clustering.

### Distinguishing homologues from analogues

Structural similarity between two sequences can be attributed to either common evolutionary ancestry (homologues) or convergent evolution (analogues). We investigated the association between cluster members, computed by our pipeline on the basis of structural similarity, and homology relationships using the ECOD database[24]. ECOD is a hierarchical domain database that describes the evolutionary relationships between pairs of protein domains. Its hierarchical levels from root to leaf are classified as: A-group (same architecture), X-group (possible homology), H-group (homology), T-group (topology) and F-group (sequence similarity). Analogues are expected to occur between members of different X-groups, whereas homologues should be found within the H-group.

For our benchmark, we downloaded the ECOD (F99 v.20230309) PDB database and applied the same MMseqs2 and Foldseek clustering procedure used for the AFDB. We conducted an ECOD cluster purity analysis on all non-singleton clusters by measuring the pairwise cluster member consistency at different hierarchy levels. The analysis revealed high average consistency rates of 99.6%, 98.6%, 97.4%, 96.8% and 72.8% for ECOD's A-group, X-group, H-group, T-group and F-group, respectively. This indicates an effective clustering of homologous proteins, demonstrating a nearly exclusive distinction between homologues and analogues. The high level of consistency in our clustering is mainly attributed to the stringent $E$-value of $10^{-2}$; when raising it to 10, the consistencies decrease to 69.7%, 55.7%, 53.3%, 51.9% and 36.6%, respectively. A similar result was observed using the MALISAM database[45], a single-domain database of analogous protein domains. When clustering the 260 protein structures within the MALISAM database with Foldseek's default parameters, no clustering of analogues occurs. However, if we increase the $E$-value threshold, we begin to form clusters containing analogues.

### Cluster purity analysis

To assess cluster purity, we followed a two-step approach. First, we calculated the average LDDT and TM score per cluster to assess the structural similarity. For this, we aligned the representative to the cluster members using the structurealign -e INF -a module in Foldseek and reported the alignment LDDT and TM score using --format-output lddt,alntmscore. For each cluster we computed the mean illustrated in Fig. 1c.

Second, we evaluated the Pfam consistency of each cluster by using Pfam labels obtained from UniProtKB. We took into account only the clusters that have at least two sequences with Pfam annotations and we calculated the fraction of correctly covered Pfam domains for all Pfam sequence pairs ignoring self-comparison. We define true positives as a pair of Pfam domains belonging to the same clan. For each pair, we computed the consistency scores by true-positive count divided by the count of Pfams in the reference sequence. Finally, we computed the mean overall pair scores. This approach enabled us to determine the proportion of sequences within a given cluster that shared the same Pfam annotation.

Finally, we also calculated the EC number consistency of each cluster. EC numbers were extracted from UniProtKB. The EC consistency was evaluated similarly to the Pfam consistency but was done four times according to the four classes of the EC number. We considered only the clusters with at least two sequences that have EC annotations. At each class of the EC number, the annotation without any code at the class was ignored. For each pair as the Pfam consistency, the consistency scores were computed by the true-positive count divided by the number of ECs in the sequences in the pair avoiding self-comparison. The scores were finally computed to the mean overall pair scores.

### Dark clusters and LCA

To eliminate clusters similar to previously known experimental structures, we conducted a search using Foldseek against the PDB (v.2022-10-14) for each cluster representative, with an $E$-value threshold of 0.1. We then excluded clusters annotated with Pfam domains by searching the cluster representatives using MMseqs2 with parameters -s 7.5 --max-seqs 100000 -e 0.001 against the Pfam database. Finally, we removed clusters with members annotated with Pfam or TIGRFAM20 annotations in the UniProt/TrEMBL and SwissProt database. To determine the LCA of each cluster, we used the lca module in MMseqs2 (ref. 46) ignoring the two taxa (1) 12,908 unclassified sequences and (2) 28,384 other sequences. We visualized the LCA results using a Sankey plot generated by Pavian[47].

### Prediction of functions and pockets

We predicted small-molecule-binding sites for representative dark cluster members by adapting a previously described approach[9]. We used AutoSite to predict pockets[48], and selected pockets with an AutoSite empirical composite score of >60 and mean pocket residue pLDDT of >90 for additional analyses. To assign putative function and predict catalytic residues, we used DeepFRI[49] to predict enriched GO/EC terms and residue-level saliency weights across available GO/EC categories (BP, CC, EC, MF). Pocket and functional predictions were then visually examined using a web app (Data Availability).

### Domain prediction from local alignments

First, we filtered out low-scoring Foldseek hits using an $E$-value of $10^{-3}$ as the threshold. We defined potential domain boundary positions for each protein sequence by clustering start–stop positions (hierarchical clustering, height parameter of 250 to establish clusters). Predicted domains were then linked to others on the basis of structural similarities, retaining the highest scores when duplicates were found. The resulting network was then trimmed excluding connections with $E$-value higher than $10^{-5}$, predicted domains with more than 350 amino acids and connected components with less than 5 nodes. We applied graph-based clustering (walktrap, 6 steps), keeping communities with at least 5 members. Each predicted domain inside the selected communities was annotated using Pfam-A regions mapped to UniProt identifiers (v.35.0), more than 75% of the Pfam domain has to overlap

with the predicted domain. We calculated inside each community the frequency of Pfam annotations and defined them on the basis of the highest one. Owing to its size, we decided to keep out of the following analysis one community with 152,959 structures (group ID 1;1, see supplementary files at https://cluster.foldseek.com/). We connected the remaining communities on the basis of the structure similarities, allowing connections with a $P < 10^{-3}$.

## Web server

We developed a web server to allow for user-friendly exploration of clusters, their members and related similar clusters. The server was implemented using a REST-based client-server architecture, with a VueJS front-end and a NodeJS back-end. The clustering-related information is accessed through an SQLite database and information related to individual structures through Foldseek compatible databases through a C++-based NodeJS-extension for fast read-in and search. Similar to the Foldseek webserver, we used NGL[50] to visualize structures and WebAssembly-based versions of PULCHRA[51] to restore full protein structures from our stored C-alpha traces and TM-align for pairwise structure alignments of cluster members to their representatives. To visualize the taxonomic distribution, we implemented Sankey diagrams inspired by Pavian. Clusters can be found through member UniProt accessions, through a Foldseek search to similar clusters or by searching for GO terms. Individual cluster members can be further explored with links to UniProt, the Foldseek webserver and the UniProt3D Atlas[52].

## Reporting summary

Further information on research design is available in the Nature Portfolio Reporting Summary linked to this article.

## Data availability

Clustering data are freely and publicly available (CC-BY) online (https://cluster.foldseek.com/). All data generated and used for the analyses can be downloaded online (https://afdb-cluster.steinegerlab.workers.dev). AlphaFold database v.3 (https://alphafold.ebi.ac.uk/) was used for the analysis and is currently available at gs://public-datasets-deepmind-alphafold. For the analysis, we used Pfam v.34.0 (https://ftp.ebi.ac.uk/pub/databases/Pfam/releases/Pfam34.0), PDB (14 October 2022; https://www.rcsb.org), UniProt TrEMBL 2022_03 (https://ftp.ebi.ac.uk/pub/databases/uniprot/), SwissProt 2022_03 (https://ftp.ebi.ac.uk/pub/databases/uniprot/), ECOD 20230309 (http://prodata.swmed.edu/ecod/) and the MALISAM (http://prodata.swmed.edu/malisam/) database.

## Code availability

The structural clustering method is available at https://foldseek.com/, is implemented in Foldseek v.4.645b789 and is available as free and open-source software (GPLv3). MMseqs2/Linclust v.14.7e284 is available online (https://mmseqs.com/). The cluster analysis was performed using goatools v.1.2.4 (https://github.com/tanghaibao/goatools), DeepFRI v.0.0.1 for GO predictions (https://github.com/flatironinstitute/DeepFRI) and ColabFold v.1.5.2 for structure prediction (https://colabfold.com). For plotting, Python v.3.10.6 (https://www.python.org/), Matplotlib v.3.6.2 (https://matplotlib.org/), seaborn v.0.12.2 (https://github.com/mwaskom/seaborn), ChimeraX v.1.5 (https://www.cgl.ucsf.edu/chimerax/), Pavian commit: cd2f21 (https://fbreitwieser.shinyapps.io/pavian/) and pandas v.1.5.2 (https://github.com/pandas-dev/pandas) were used.

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

**Acknowledgements** M.S. acknowledges support by the National Research Foundation of Korea grants (2020M3-A9G7-103933, 2021-R1C1-C102065 and 2021-M3A9-I4021220), Samsung DS research fund and the Creative-Pioneering Researchers Program through Seoul National University. P.B. is supported by the Helmut Horten Stiftung and the ETH Zurich Foundation. M.M. acknowledges support by the National Research Foundation of Korea (grant RS-2023-00250470). We thank E. L. Karin for revising text and H. Kim for improving the visualization.

**Author contributions** I.B.-H., J.Y., P.B. and M.S. designed the research. I.B.-H., J.Y., J.J., P.B. and M.S. performed analysis. J.Y., M.M. and C.L.M.G. developed the webserver, J.J. developed the dark cluster website. T.W. helped with AIM2 and gasdermin analysis. M.S. developed Foldseek cluster. M.V. and S.V. helped with AlphaFold-related questions. I.B.-H., J.Y., J.J., M.M., P.B. and M.S. wrote the manuscript.

**Funding** Open access funding provided by Swiss Federal Institute of Technology Zurich.

**Additional information**
**Correspondence and requests for materials** should be addressed to Pedro Beltrao or Martin Steinegger.

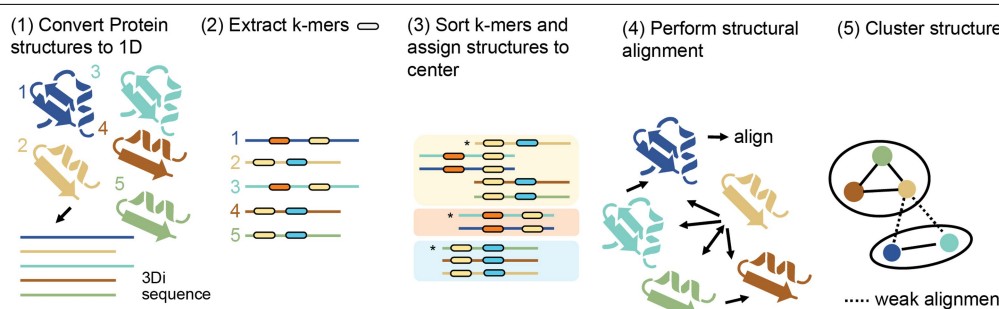

**Extended Data Fig. 1 | The five-step clustering pipeline for efficiently clustering millions of protein structures using Foldseek's 3Di alphabet.** (1) Protein structures are converted to 3Di sequences and processed through the Linclust workflow. (2) For each sequence, 300 min-hasing k-mers are extracted and sorted. (3) The longest structure is assigned to be the centre of each k-mer cluster. (4) Structural alignment is performed in two stages: first an ungapped alignment based on shared diagonal information is performed, hits are pre-clustered and second the remaining sequences are aligned using Foldseek's structural Smith-Waterman. (5) The remaining structures meeting alignment criteria are clustered using MMseqs2's clustering module. After the Linclust step the centroids are successively clustered by three cascaded steps of prefiltering, structural Smith-Waterman alignment and clustering using Foldseek's search.

a

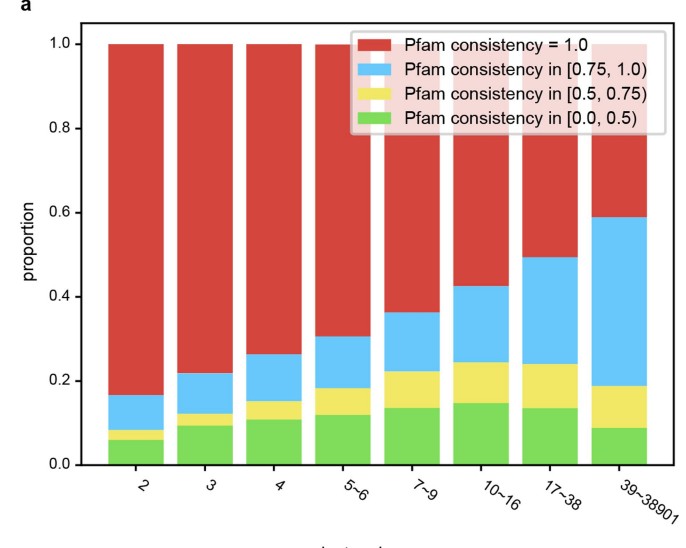

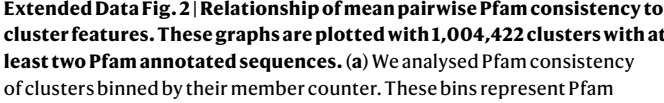

cluster size range

b

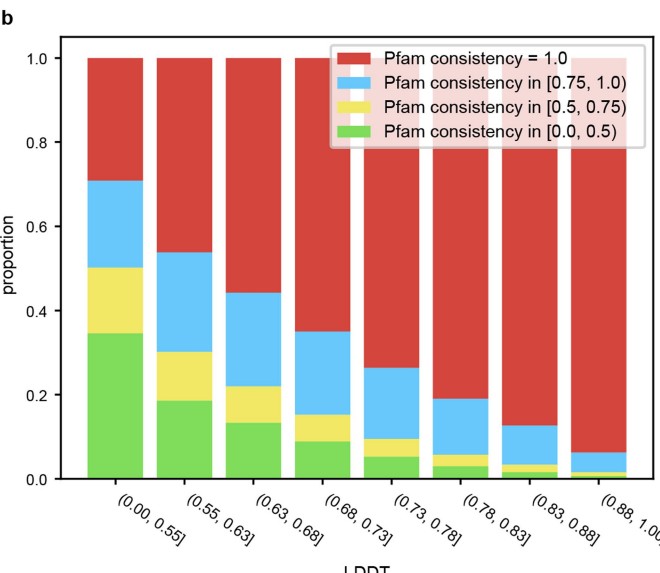

LDDT

**Extended Data Fig. 2 | Relationship of mean pairwise Pfam consistency to cluster features. These graphs are plotted with 1,004,422 clusters with at least two Pfam annotated sequences.** (**a**) We analysed Pfam consistency of clusters binned by their member counter. These bins represent Pfam annotated non-singleton clusters at rates of 19.2%, 13.5%, 9.5%, 12.6%, 11.0%, 12.4%, 11.8% and 10.0% from left to right, respectively. (**b**) We analysed Pfam consistency of clusters binned by their LDDT of each cluster. These bins represent Pfam annotated non-singleton clusters equally.

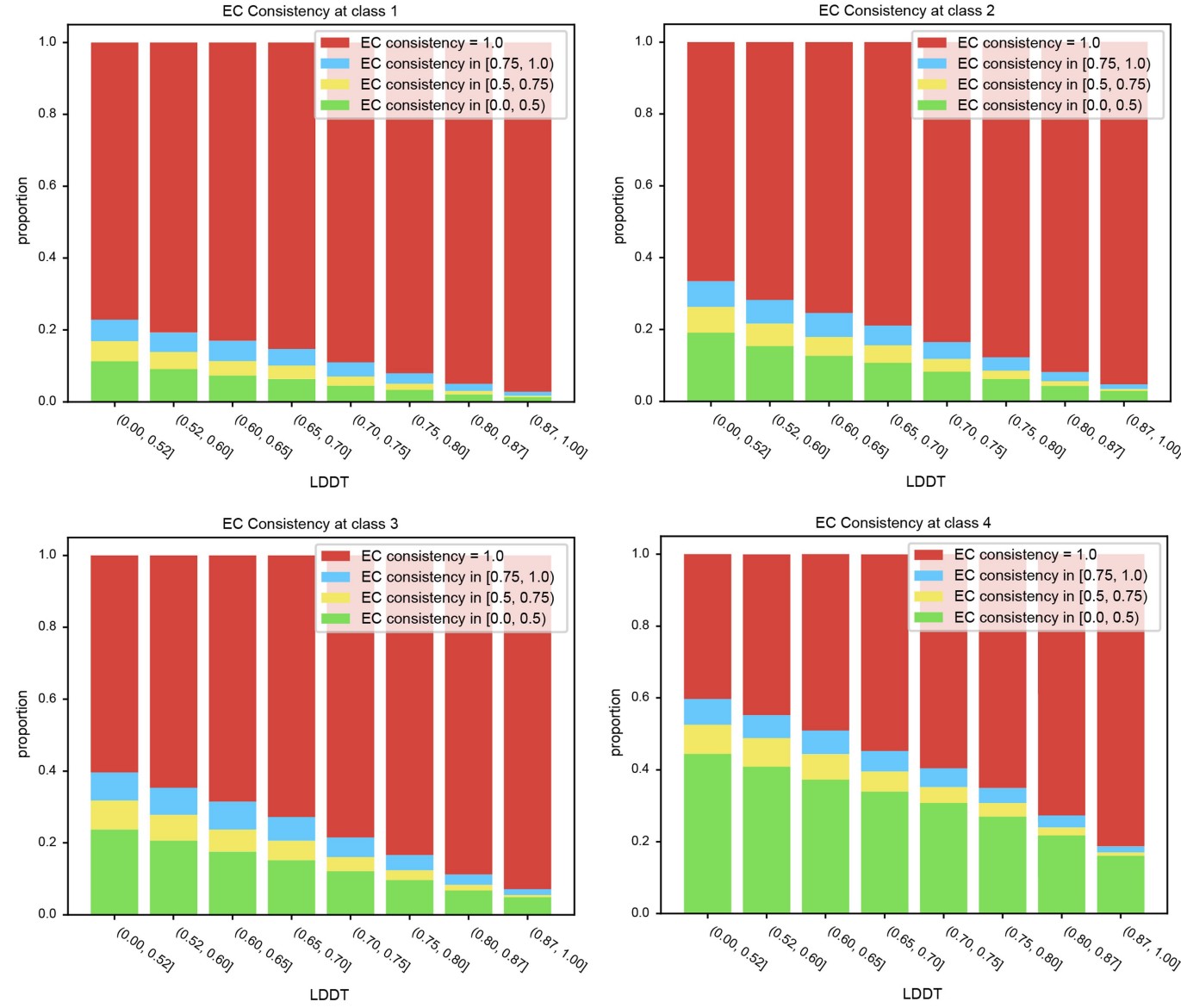

**Extended Data Fig. 3 | Relationship of mean pairwise EC number consistency to LDDT of cluster.** These graphs are plotted with 113,287 clusters with at least two Enzyme Commission number annotated sequences. Each panel describes EC consistency compared at 1 to 4 classes. Each bin in a panel represents EC annotated non-singleton clusters equally.

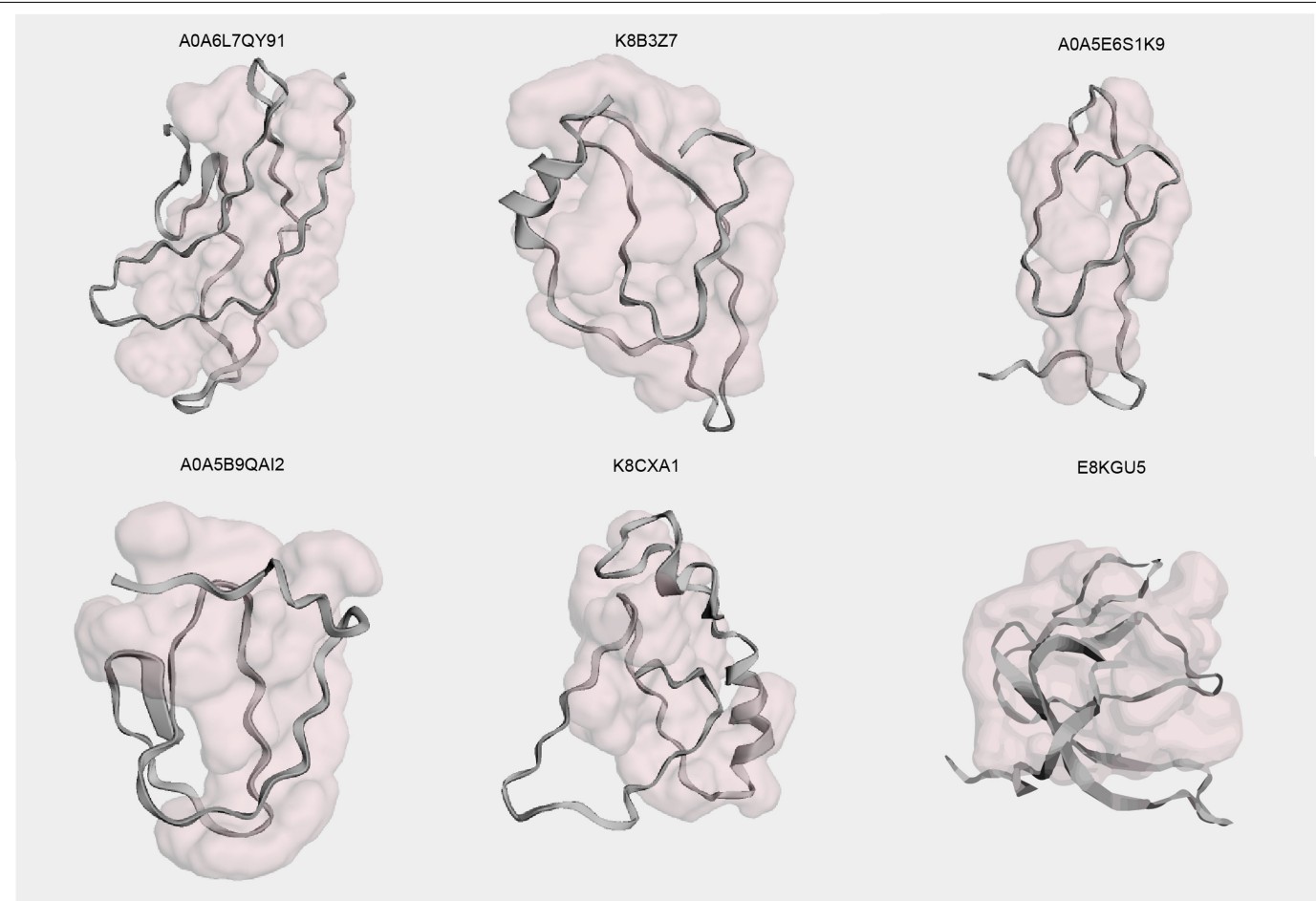

**Extended Data Fig. 4 | Examples of non-compact AlphaFold2 predicted structures.** Examples of representative structures of clusters without annotations having pLDDT>90 and a predicted pocket covering over 80% of the residues of the structure.

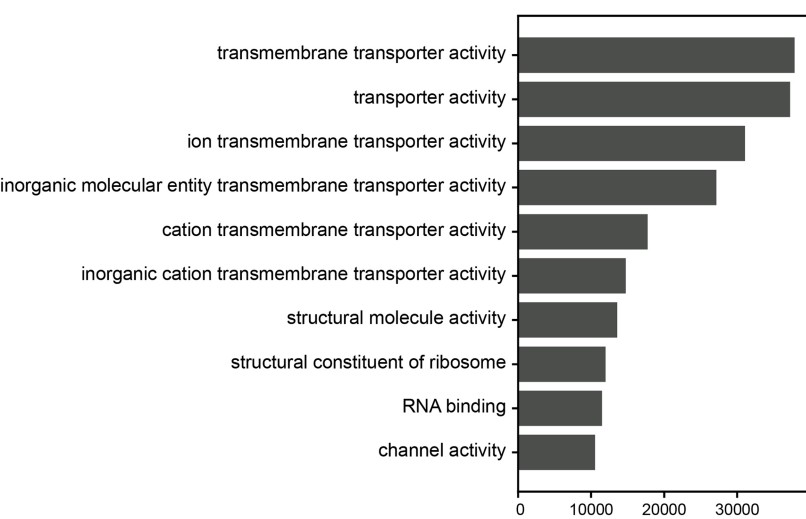

Top predicted molecular functions in 712k dark clusters with prediction score > 0.5

**Extended Data Fig. 5 | Top predicted molecular functions in all 712k dark clusters with DeepFRI scores greater than 0.5.** The graph displays the most frequent molecular functions predicted by DeepFRI with prediction scores above 0.5 across all 712k dark clusters, highlighting the prevalence of the keyword "transmembrane". Only 98,882 (13.9%) out of the 712K have a prediction score greater than 0.5.

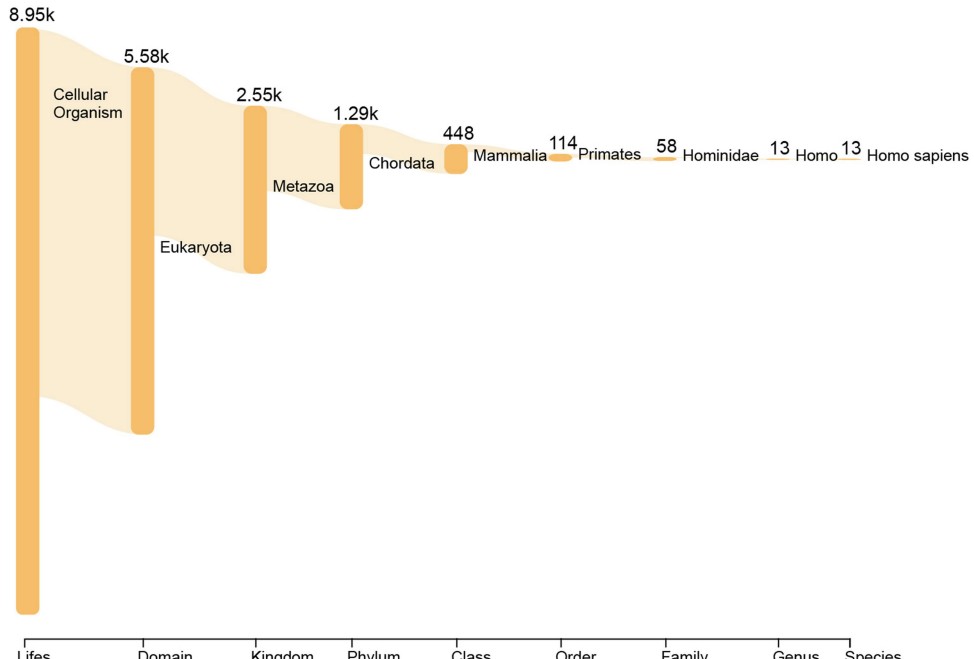

**Extended Data Fig. 6 | LCA plot of the clusters that contain Homo Sapiens proteins.** Lowest common ancestor Sankey plot generated by Pavian for all clusters containing human proteins.

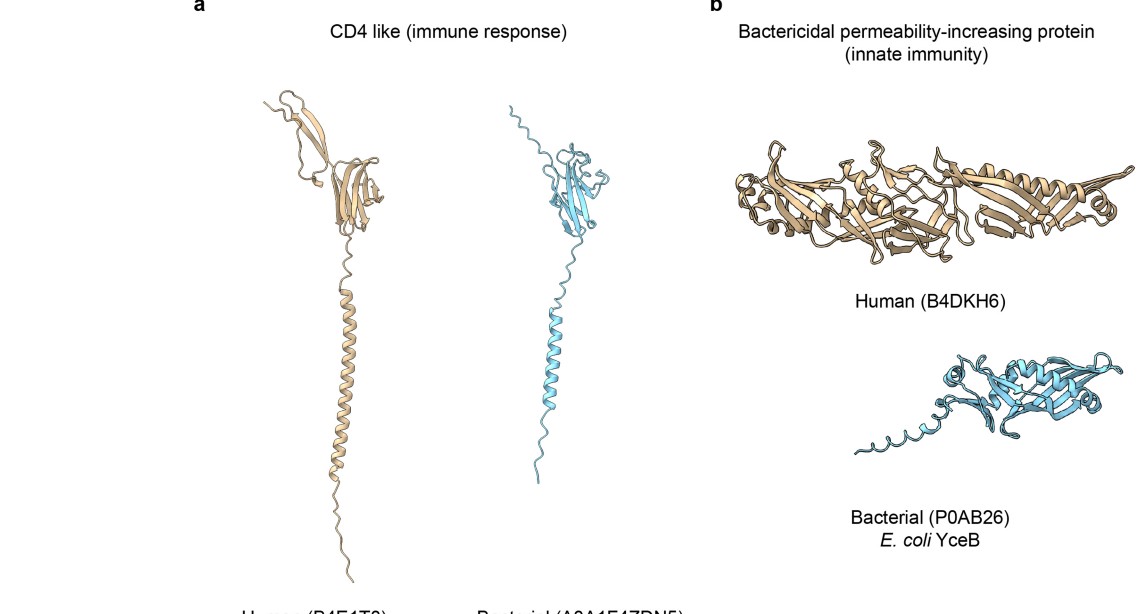

**a**

CD4 like (immune response)

Human (B4E1T0)  Bacterial (A0A1F4ZDN5)

**b**

Bactericidal permeability-increasing protein
(innate immunity)

Human (B4DKH6)

Bacterial (P0AB26)
*E. coli* YceB

**Extended Data Fig. 7 | Additional examples of human related proteins in structural clusters with representatives or partial matches in bacterial species.** (**a**) We found bacterial structures related to the human CD4 like protein B4E1T0. The human protein (B4E1T0) has 3 Pfams - PF05790, PF09191, PF12104. Those Pfams are specific to Eukaryotes only. In contrast, the bacterial protein (A0A1F4ZDN5) has no Pfam annotation. (**b**) The human protein (B4DKH6) is a bactericidal permeability-increasing protein found in humans. The *E. coli* protein (P0AB26) has a similar structure to the human protein, contains a Pfam domain of unknown function (DUF) and its structure is also experimentally determined (PDB: 3l6i B).

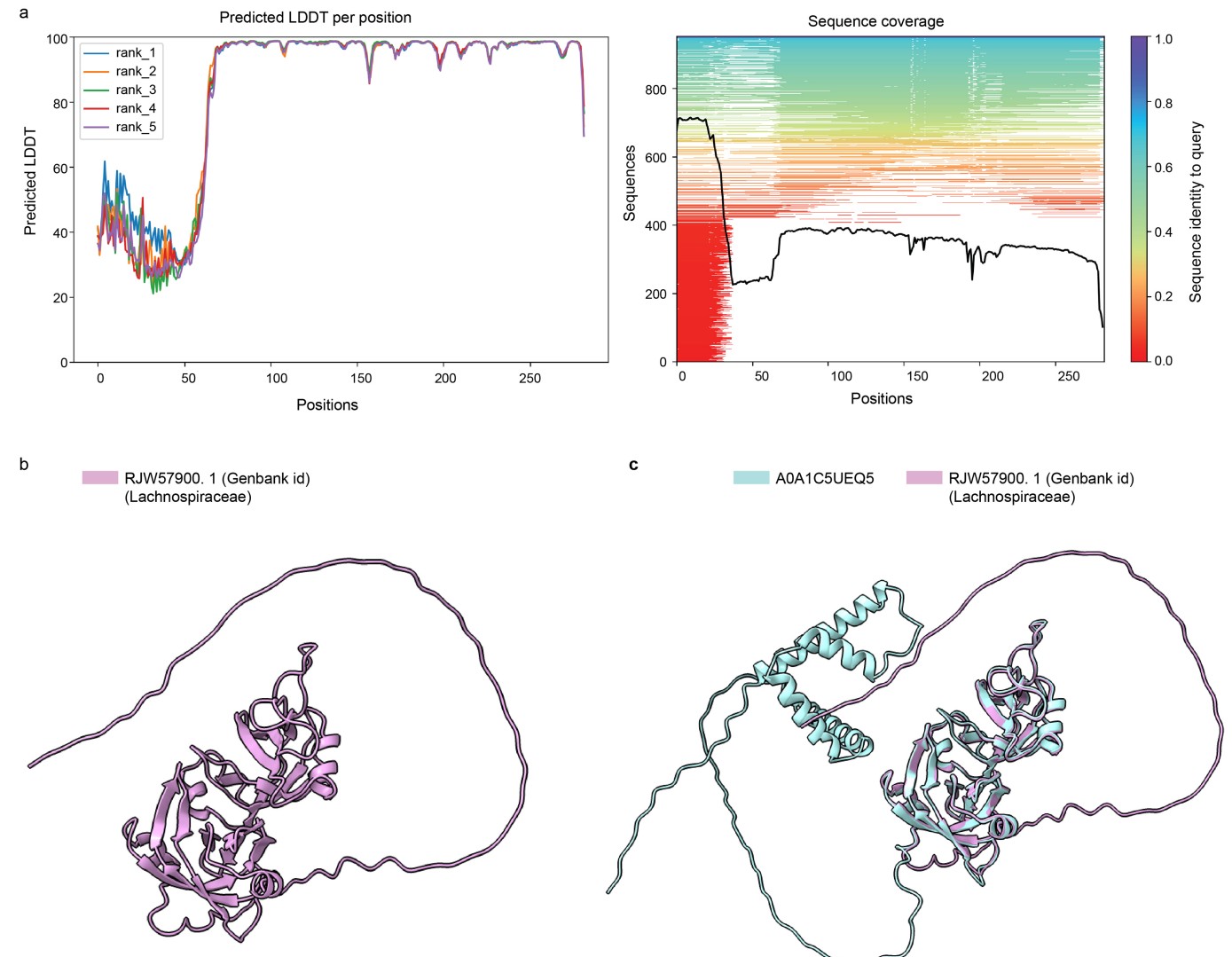

**Extended Data Fig. 8 | Comparison of predicted structures of homologous proteins: _Lachnospiraceae_ bacterium to _Clostridium_. (a)** pLDDT and multiple-sequence-alignment coverage output produced by ColabFold for the prediction of the protein sequence of _Lachnospiraceae_. (**b**) The predicted structure of RJW57900.1. (C) Superposition of the _Clostridium_ protein structure with _Lachnospiraceae_ with the DNA binding domain being well superposable.

# Reporting Summary

## Statistics

For all statistical analyses, confirm that the following items are present in the figure legend, table legend, main text, or Methods section.

| n/a | Confirmed | |
|---|---|---|
| ☐ | ☒ | The exact sample size (*n*) for each experimental group/condition, given as a discrete number and unit of measurement |
| ☒ | ☐ | A statement on whether measurements were taken from distinct samples or whether the same sample was measured repeatedly |
| ☐ | ☒ | The statistical test(s) used AND whether they are one- or two-sided<br>*Only common tests should be described solely by name; describe more complex techniques in the Methods section.* |
| ☒ | ☐ | A description of all covariates tested |
| ☐ | ☒ | A description of any assumptions or corrections, such as tests of normality and adjustment for multiple comparisons |
| ☐ | ☒ | A full description of the statistical parameters including central tendency (e.g. means) or other basic estimates (e.g. regression coefficient) AND variation (e.g. standard deviation) or associated estimates of uncertainty (e.g. confidence intervals) |
| ☐ | ☒ | For null hypothesis testing, the test statistic (e.g. *F*, *t*, *r*) with confidence intervals, effect sizes, degrees of freedom and *P* value noted<br>*Give P values as exact values whenever suitable.* |
| ☒ | ☐ | For Bayesian analysis, information on the choice of priors and Markov chain Monte Carlo settings |
| ☒ | ☐ | For hierarchical and complex designs, identification of the appropriate level for tests and full reporting of outcomes |
| ☒ | ☐ | Estimates of effect sizes (e.g. Cohen's *d*, Pearson's *r*), indicating how they were calculated |

*Our web collection on statistics for biologists contains articles on many of the points above.*

## Software and code

Policy information about availability of computer code

| Data collection | The structural clustering method is available at foldseek.com, is is implemented in Foldseek v4.645b789 and available as free and open-source software (GPLv3). MMseqs2/Linclust v14.7e284 is available at mmseqs.com. |
|---|---|
| Data analysis | The cluster analysis was performed using goatools v1.2.4 (https://github.com/tanghaibao/goatools), DeepFRI v0.0.1 for GO predictions (https://github.com/flatironinstitute/DeepFRI) ColabFold v1.5.2 for structure prediction (https://colabfold.com). For plotting Python v3.10.6 (https://www.python.org/), Matplotlib v3.6.2 (https://matplotlib.org/), seaborn v0.12.2 (https://github.com/mwaskom/seaborn), ChimeraX v1.5 (https://www.cgl.ucsf.edu/chimerax/), Pavian commit: cd2f21 (https://fbreitwieser.shinyapps.io/pavian/), pandas v1.5.2 (https://github.com/pandas-dev/pandas) was used. |

For manuscripts utilizing custom algorithms or software that are central to the research but not yet described in published literature, software must be made available to editors and reviewers. We strongly encourage code deposition in a community repository (e.g. GitHub). See the Nature Portfolio guidelines for submitting code & software for further information.

## Data

Policy information about availability of data

  All manuscripts must include a data availability statement. This statement should provide the following information, where applicable:
- Accession codes, unique identifiers, or web links for publicly available datasets
- A description of any restrictions on data availability
- For clinical datasets or third party data, please ensure that the statement adheres to our policy

Clustering data is freely and publicly available (CC-BY) at cluster.foldseek.com
All data generated and used for the analyses can be downloaded: https://afdb-cluster.steineggerlab.workers.dev

AlphaFold database v3 (https://alphafold.ebi.ac.uk/) was used for the analysis and is currently available at gs://public-datasets-deepmind-alphafold. For the analysis we used Pfam 34.0 (https://ftp.ebi.ac.uk/pub/databases/Pfam/releases/Pfam34.0), PDB (Oct 14th, 2022. https://www.rcsb.org), UniProt TrEMBL 2022_03 (https://ftp.ebi.ac.uk/pub/databases/uniprot/), SwissProt 2022_03 (https://ftp.ebi.ac.uk/pub/databases/uniprot/), ECOD 20230309 (http://prodata.swmed.edu/ecod/) and the MALISAM (http://prodata.swmed.edu/malisam/) database.

## Research involving human participants, their data, or biological material

Policy information about studies with human participants or human data. See also policy information about sex, gender (identity/presentation), and sexual orientation and race, ethnicity and racism.

| Reporting on sex and gender | Not applicable. The terms - sex and gender - were not used in the paper. |
|---|---|
| Reporting on race, ethnicity, or other socially relevant groupings | Not applicable. No social analysis is done in the paper. |
| Population characteristics | Not applicable. No human participant is included in the analysis |
| Recruitment | Not applicable. No participant is included in the analysis |
| Ethics oversight | Not applicable. |

Note that full information on the approval of the study protocol must also be provided in the manuscript.

# Field-specific reporting

Please select the one below that is the best fit for your research. If you are not sure, read the appropriate sections before making your selection.

☒ Life sciences  ☐ Behavioural & social sciences  ☐ Ecological, evolutionary & environmental sciences

For a reference copy of the document with all sections, see nature.com/documents/nr-reporting-summary-flat.pdf

# Life sciences study design

All studies must disclose on these points even when the disclosure is negative.

| Sample size | The entire size of AlphaFold2 Database v3 -214,684,311 protein structures - were used. |
|---|---|
| Data exclusions | We excluded fragmented proteins, singleton clusters and redundant protein through MMseqs2 clustering. Mainly, the analyses in the paper were done with the output of the process aforementioned - 2,302,908 clusters and 30,045,247 protein entries. But those which are excluded are depicted in the paper and provided in our data website (cluster.foldseek.com) |
| Replication | Not applicable. The outputs in the paper are done by computational methods. The computational method is deterministic so the result is identical for each replicate. |
| Randomization | Not applicable. We are not comparing across groups. |
| Blinding | Not applicable. We are not comparing across groups. |

# Reporting for specific materials, systems and methods

We require information from authors about some types of materials, experimental systems and methods used in many studies. Here, indicate whether each material, system or method listed is relevant to your study. If you are not sure if a list item applies to your research, read the appropriate section before selecting a response.

## Materials & experimental systems

| n/a | Involved in the study |
|-----|----------------------|
| ☒ | ☐ Antibodies |
| ☒ | ☐ Eukaryotic cell lines |
| ☒ | ☐ Palaeontology and archaeology |
| ☒ | ☐ Animals and other organisms |
| ☒ | ☐ Clinical data |
| ☒ | ☐ Dual use research of concern |
| ☒ | ☐ Plants |

## Methods

| n/a | Involved in the study |
|-----|----------------------|
| ☒ | ☐ ChIP-seq |
| ☒ | ☐ Flow cytometry |
| ☒ | ☐ MRI-based neuroimaging |

