## [Peer Review File · Nature]

Manuscript Title: Clustering predicted structures at the scale of the known protein universe

Reviewer Comments & Author Rebuttals

Reviewer Reports on the Initial Version:

Referees' comments:

Referee #1:

A. Summary of the key results

The submitted work presents a new algorithm for clustering extremely large protein structure databases. The method was applied to 52 million representative (no more than 50% sequence identity) structures from the AFDB, a database of predicted structures obtained with AlphaFold2.

Structure-based clustering yielded >2 million clusters, each containing 2 or more structures, of which 31% are "dark", i.e. they contain mostly unannotated proteins. The authors show various analyses based on the clustering, such as the detection of distant similarities and novel domains. The "dark" clusters were also analyzed for the possibility of their enzymatic activity.

All data (clustering and annotation results) are available via an easy-to-use server. The developed software is available and well documented.

B. Originality and significance

The presented method is robust and well documented. It will certainly be widely used because of its unique ability to cluster very large protein structure datasets, such as the AFDB. The analyses performed clearly illustrate how the proposed approach can be used for different types of studies.

C. Data & methodology

Data and methods are robust and of high quality.

D. Appropriate use of statistics and treatment of uncertainties

Not applicable.

E. Conclusions: robustness, validity, reliability

The section "Bacterial remote homology of human immunity related proteins" raises an important issue of detecting homology based on structural similarity. The current narrative of the paper suggests that structural similarity is a clear indication of homology, which I believe is not always true. There are clear examples of structures that are very similar but analogous, i.e., not of common origin. Some such examples are catalogued in the MALISAM database (<http://prodata.swmed.edu/malisam/>). It is also well known that "superfolds" such as the immunoglobulin fold, the jelly roll fold, and the Rossmann fold contain pairs of proteins whose homology is disputed. Browsing the AFDB Clusters database reveals examples where clearly non-homologous proteins are grouped into a cluster because they share structurally similar domains.

Of course, the examples given by the authors may be homologous (e.g., the case of bacterial and eukaryotic histones, whose homology was proposed based on sequence searches and expert knowledge), but this is not a general rule. A similar consideration can be made for the paragraph

"Structural similarity across distantly related domain families". Searching the Pfam database using HHpred and the DUF998/PF06197 seed alignment as query gives a good hit (>97% probability) to PF10277, so the connection between these two domains seems to be the result of homology, but other connections based only on structural similarity remain in the "twilight zone".

I trust that these sections and relevant fragments of the introduction and discussion should be rewritten to avoid implying that structural similarity equals homology. It would also be great to see additional analysis and discussion of this issue. For example, one could cluster the structures from the ECOD database (<http://prodata.swmed.edu/ecod/>) and see if the resulting clusters contain domains from different architectures or X-groups.

A related question is how informative structural similarity is about function. Proteins belonging to large classes, such as the P-loop or the Rossmanns mentioned above, encompass very different functions, the similarity of which may boil down, for example, to the use of the same cofactor. I would encourage the authors to perform some kind of analysis that relates structural similarity (TM score?) to the degree of unambiguity of the structure-based functional annotation.

F. Suggested improvements: experiments, data for possible revision

At <https://cluster.foldseek.com/> the "dark enzymes prediction" link does not work ("AttributeError: This app has encountered an error"). The AlphaFold Clusters interface is useful, but some features could be added. For example, a useful feature would be a browse option to allow identification of keywords, species specific clusters, dark clusters, etc. This data is in the flat files, but it would be good to make it accessible via the server as well.

I think it would be beneficial to expand the section "Prediction of putative novel enzymes and small molecule binding proteins" a bit so that it's clear without referring to the methods. For example, the sentence "the pocket prediction led to the identification of high confidence structure predictions (pLDDT>90) that don't appear to be correct" was not immediately clear to me. I understand that this means that some of the predicted pockets (with high confidence and mean pLDDT of the constituent residues >90) cover more than 40% of the total structure, what suggests that they are wrong?

G. References

The bibliography seems to be complete, although the lack of references to ESM is surprising. Also, one would expect ESMfold and ESM-Atlas to be mentioned in the discussion, especially since it contains more structures than AFDB.

H. Clarity and context

The abstract is clear and accessible; however, it contains statements related to structure-based homology inference that I think need to be clarified (see point E).

Referee #2:

The manuscript highlights the excellent computational tools for protein sequence and structure similarity rapid searches developed by the authors and (already!!!) very widely used by computational biologists worldwide. Currently, only the methods developed by these authors can handle the vast numbers of protein models in the AF database and provide their clustering by similarity for the exploration by the research community. All the tools and data are made available, and this work is significant.

The authors have chosen a friendly format to present their recently developed computational

methods as a comprehensive application to the 200 million models in the AF database, thus providing a (nearly) comprehensive description of the protein domain space. The work is thoroughly executed, and the methods are well-described. The examples of proteins involved in innate immunity and gasdermin domains are exciting and insightful. Overall, this work is an introduction to the tools developed by the authors and gives a glimpse of the discoveries that can be made by mining the data using their tools.

My main reservation is about the presentation style selected by the authors. It is more detail-oriented than general and may be somewhat challenging to digest for the wider audience unfamiliar with the jargon in the field (the most striking example is that although LCA, a term familiar to many biologists, is defined, pLDDT is not defined anywhere!). Many sections of results seem too detailed, listing every possible number, and the major messages of these paragraphs are getting lost among these details. This writing style is certainly very good for computational and structural biologists, but I wonder if the readers outside these fields will be much enlightened by it. To significantly increase the impact of this paper, I suggest a partial rewrite to bring up the essence and present most numbers in tables or in a supplement for specialists to study. I think consultations with general biologists and asking for their opinions about what they would like to see in various main text paragraphs could be helpful. The paper is for them to read, not for the authors. The methods developed and presented here are wonderful and will be widely used; we all know that.

In summary, I would like to see the amazing ultra-fast and most useful methods highlighted in Nature, and re-wording some sections of the paper will increase its general readability and, therefore, its impact.

Author Rebuttals to Initial Comments:

Referees' comments:

Referee #1:

A. Summary of the key results

The submitted work presents a new algorithm for clustering extremely large protein structure databases. The method was applied to 52 million representative (no more than 50% sequence identity) structures from the AFDB, a database of predicted structures obtained with AlphaFold2.

Structure-based clustering yielded >2 million clusters, each containing 2 or more structures, of which 31% are "dark", i.e. they contain mostly unannotated proteins. The authors show various analyses based on the clustering, such as the detection of distant similarities and novel domains. The "dark" clusters were also analyzed for the possibility of their enzymatic activity.

All data (clustering and annotation results) are available via an easy-to-use server. The developed software is available and well documented.

Thank you for raising important points as well as the encouraging words.

B. Originality and significance

The presented method is robust and well documented. It will certainly be widely used because of its unique ability to cluster very large protein structure datasets, such as the AFDB. The analyses performed clearly illustrate how the proposed approach can be used for different types of studies.

C. Data & methodology

Data and methods are robust and of high quality.

D. Appropriate use of statistics and treatment of uncertainties

Not applicable.

E. Conclusions: robustness, validity, reliability

The section "Bacterial remote homology of human immunity related proteins" raises an important issue of detecting homology based on structural similarity. The current narrative of the paper suggests that structural similarity is a clear indication of homology, which I believe is not always true. There are clear examples of structures that are very similar but analogous, i.e., not of common origin. Some such examples are catalogued in the MALISAM database (<http://prodata.swmed.edu/malisam/>). It is also well known that "superfolds" such as the immunoglobulin fold, the jelly roll fold, and the Rossmann fold contain pairs of proteins whose homology is disputed. Browsing the AFDB Clusters database reveals examples where clearly non-homologous proteins are grouped into a cluster because they share structurally similar domains.

Of course, the examples given by the authors may be homologous (e.g., the case of bacterial and eukaryotic histones, whose homology was proposed based on sequence searches and expert knowledge), but this is not a general rule. A similar consideration can be made for the paragraph "Structural similarity across distantly related domain families". Searching the Pfam database using HHpred and the DUF998/PF06197 seed alignment as query gives a good hit (>97% probability) to PF10277, so the connection between these two domains seems to be the result of homology, but other connections based only on structural similarity remain in the "twilight zone".

I trust that these sections and relevant fragments of the introduction and discussion should be rewritten to avoid implying that structural similarity equals homology. It would also be great to see additional analysis and discussion of this issue. For example, one could cluster the structures from the ECOD database (<http://prodata.swmed.edu/ecod/>) and see if the resulting clusters contain domains from different architectures or X-groups.

We appreciate the point raised here and fully agree that structural similarity does not imply a common evolutionary origin. We have revised the manuscript to remove or reduce explicit mentions of homology, either replacing the term with structural similarity or explicitly saying that these are predicted homology. We have also added this point to the discussion section to be clear about these differences, including references to the database resources listed by the reviewer.

Thank you for suggesting the ECOD-based benchmark. We added an ECOD F99 database benchmark, by clustering it following the same protocol proposed in the manuscript. Using the cut-offs that we describe in the manuscript, we find high average consistency rates of 99.6%, 98.6%, 97.4%, 96.8%, and 72.8% for ECOD's A-group, X-group, H-group, T-group, and F-group, respectively. This indicates an effective clustering of homologous proteins, demonstrating nearly exclusive distinction between homologs and analogs. We also clustered the MALISAM structure and the analogs pairs did not cluster together using our default E-value. We think these results suggest that the clustering cut-offs used here will preferentially group together homologs instead of analogs. It is useful to keep in mind that the clustering cut-off was at 90% length of sequences and fairly stringent E-value. In fact, when raising the E-value from 10^{-2} to 10 the consistencies decrease to 69.7%, 55.7%, 53.3%, 51.9% and 36.6%, respectively. Overall, we think our analysis and methods will open the door to study convergent evolution of protein structure more broadly, but quite a substantial and dedicated effort will need to be done to address this question properly.

A related question is how informative structural similarity is about function. Proteins belonging to large classes, such as the P-loop or the Rossmanns mentioned above, encompass very different functions, the similarity of which may boil down, for example, to the use of the same cofactor. I would encourage the authors to perform some kind of analysis that relates structural similarity (TM score?) to the degree of unambiguity of the structure-based functional annotation.

This is a great suggestion. We added two analyses where we compared the relationship of average LDDT per cluster compared to PFAM or EC annotation consistency (see plots below). We picked LDDT over TM-score to account for the flexibility of domains. As expected, the higher the structural similarity, the higher the consistency.

F. Suggested improvements: experiments, data for possible revision

At <https://cluster.foldseek.com/> the "dark enzymes prediction" link does not work ("AttributeError: This app has encountered an error"). The AlphaFold Clusters interface is useful, but some features could be added. For example, a useful feature would be a browse option to allow identification of keywords, species specific clusters, dark clusters, etc. This data is in the flat files, but it would be good to make it accessible via the server as well.

Thank you for testing the website. We fixed the issue related to the "dark enzyme prediction" website. Despite this there is a chance that the cloud provider introduces a breaking change. Therefore, we have proactively built a workaround. We have written a guide on how to run the website locally, effectively ensuring uninterrupted access to the dark enzyme predictions.

We agree strongly that improving the cluster search capabilities is important. To make browsing easier, we've implemented a variety of new search functionalities. Users now have the ability to navigate through clusters using Gene Ontology (GO) terms, taxonomy, darkness, length, cluster size, among others. We deployed the features on cluster-dev.foldseek.com, and we are going to release them within the next weeks.

I think it would be beneficial to expand the section "Prediction of putative novel enzymes and small molecule binding proteins" a bit so that it's clear without referring to the methods. For example, the sentence "the pocket prediction led to the identification of high confidence structure predictions (pLDDT>90) that don't appear to be correct" was not immediately clear to me. I understand that this means that some of the predicted pockets (with high confidence and mean pLDDT of the constituent residues >90) cover more than 40% of the total structure, what suggests that they are wrong?

As per the suggestion of the reviewer, we have revised this sentence to explain better to the reader what we mean with this observation: "From 1723 pockets, 559 (32%) encompass more than 40% of the total protein sequence (examples are shown in SFig. 2), indicating that the predicted structure is not compact. Manual inspection of these structures (see examples in SFig. 2) confirmed this lack of compactness and secondary structural elements. We hypothesize that several of these are likely to be incorrect predictions."

We agree that we cannot rule out that these predictions are correct. Looking at several examples of these predictions, they don't look like well-ordered folded structures. Often the high confidence predictions are typically associated with such well-ordered structures. If these proteins are not well ordered then it is also unlikely that they will be in single conformation. As such, the predicted pocket is not likely to be real, and these wouldn't be enzymes.

G. References

The bibliography seems to be complete, although the lack of references to ESM is surprising. Also, one would expect ESMfold and ESM-Atlas to be mentioned in the discussion, especially since it contains more structures than AFDB

Very good point, we agree with the reviewer and this was indeed an oversight. We added the reference in the discussion section, exactly emphasizing the larger size of ESM-Atlas.

H. Clarity and context

The abstract is clear and accessible; however, it contains statements related to structure-based homology inference that I think need to be clarified (see point E).

See the answer on top, we have revised these.

Referee #2:

The manuscript highlights the excellent computational tools for protein sequence and structure similarity rapid searches developed by the authors and (already!!!) very widely used by computational biologists worldwide. Currently, only the methods developed by these authors can handle the vast numbers of protein models in the AF database and provide their clustering by similarity for the exploration by the research community. All the tools and data are made available, and this work is significant.

Thank you so much for your valuable feedback as well as the encouraging words.

The authors have chosen a friendly format to present their recently developed computational methods as a comprehensive application to the 200 million models in the AF database, thus providing a (nearly) comprehensive description of the protein domain space. The work is thoroughly executed, and the methods are well-described. The examples of proteins involved in innate immunity and gasdermin domains are exciting and insightful. Overall, this work is an introduction to the tools developed by the authors and gives a glimpse of the discoveries that can be made by mining the data using their tools.

My main reservation is about the presentation style selected by the authors. It is more detail-oriented than general and may be somewhat challenging to digest for the wider audience unfamiliar with the jargon in the field (the most striking example is that although LCA, a term familiar to many biologists, is defined, pLDDT is not defined anywhere!). Many sections of results seem too detailed, listing every possible number, and the major messages of these paragraphs are getting lost among these details. This writing style is certainly very good for computational and structural biologists, but I wonder if the readers outside these fields will be much enlightened by it. To significantly increase the impact of this paper, I suggest a partial rewrite to bring up the essence and present most numbers in tables or in a supplement for specialists to study. I think consultations with general biologists and asking for their opinions about what they would like to see in various main text paragraphs could be helpful. The paper is for them to read, not for the authors. The methods developed and presented here are wonderful and will be widely used; we all know that.

We understand the concern, and we agree that for a general audience, the text may have some unnecessary jargon and quite a lot of details. We have tried to revise the text, particularly towards the start, where many explicit numbers were given, perhaps in a way that made it harder to read. We also defined some of the acronyms that were not previously defined before. We hope the paper is now easier to read.

In summary, I would like to see the amazing ultra-fast and most useful methods highlighted in Nature, and re-wording some sections of the paper will increase its general readability and, therefore, its impact.

Again thank you for the helpful comments.